# Factor associated with anthropometric failure among under-five Bengali children: A comparative study between Bangladesh and India

**Ramendra Nath Kundu[1], Md. Golam Hossain** **[2]\*, Md. Ahshanul Haque[3], Subir Biswas[1], Md. Monimul Huq[2], Md. Kamal Pasa[4], Md. Sabiruzzaman[2], Premananda Bharati[5]**

1 Department of Anthropology, West Bengal State University, Kolkata, West Bengal, India, 2 Department of Statistics, University of Rajshahi, Rajshahi, Bangladesh, 3 Nutrition and Clinical Services Division, icddr,b, Dhaka, Bangladesh, 4 Department of Anthropology, University of Rajshahi, Rajshahi, Bangladesh, 5 Biological Anthropology Unit, Indian Statistical Institute, Kolkata, West Bengal, India

\* hossain95@yahoo.com

## Abstract

### Background

Child undernutrition is a burden and the leading cause of child mortality in low-and middle-income countries like Bangladesh and India. Currently, this issue is a matter of great concern, inasmuch as achieving the Sustainable Development Goals (SDGs). The study intends to determine the factors of child undernutrition using a single composite index of anthropometric failure (CIAF) among the Bengali population.

### Methods

Unit level data on 14055 under 5 children were extracted from the Bangladesh Demographic and Health Survey 2017–18 (BDHS) and the 4th National Family Health Survey of India (NFHS-4). To understand child undernutrition and generate CIAF, data on height-for-age (stunting), weight-for-height (wasting), and weight-for-age (underweight) were used by WHO guidelines. These three undernutrition indicators were combined into a single under-nutrition indicator called anthropometric failure (anth-failure) using the CIAF concept. Explanatory factors of anth-failure included data on maternal health, socio-demographic and birth-related variables. Differences of frequency were determined by Z-proportional and Chi-square tests; predictors of anth-failure were determined by binary logistic regression. Cut off point of p-value was taken as 0.05 to test the significance.

### Results

Inter-country disparities were revealed, about half of Bengali children in India and two-fifths in Bangladesh being prone to anth-failure. Stunting and underweight were more prevalent in both countries than wasting. Maternal undernutrition, lack of maternal education, and poor wealth index were common factors of anth-failure for both countries. Children in Bangladesh

**Data Availability Statement:** Supporting data for this study are available in Demographic and Health Surveys (DHS) website and publicly available. The

NFHS-4, 2015-2016 and the BDHS, 2017-2018 datasets are freely available at https://dhsprogram.com/data/dataset/India_Standard-DHS_2015.cfm?flag=0 and https://dhsprogram.com/data/dataset/Bangladesh_Standard-DHS_2017.cfm?flag=0 respectively.

**Funding:** The author(s) received no specific funding for this work.

**Competing interests:** The authors have declared that no competing interests exist.

**Abbreviations:** AF, Anthropometric Failure; AOR, Adjusted Odds Ratio; BD, Bangladesh; BDHS, Bangladesh Demographic and Health Survey; BMI, Body Mass Index; CI, Confidence Interval; CIAF, Composite Index of Anthropometric Failure; DHS, Demographic and Health Surveys; IIPS, International Institute for Population Sciences; IN, India; LMICs, Low-and Middle-Income Countries; NAF, No Anthropometric Failure; NFHS, National Family Health Survey; NIPORT, National Institute of Population Research and Training; SD, Standard Deviation; SDGs, Sustainable Development Goals; SPSS, Statistical Package for the Social Sciences; UN, United Nations; VIF, Variance Inflation Factor; WHO, World Health Organization.

developed anth-failure after the end of breastfeeding period, indicating a lack of nutritious food. Lack of antenatal care was another significant factor in Bangladesh. In India, the first child suffered from anth-failure due to lack of maternal education.

## Conclusions

This study provides a better understanding of multifactorial impact on child undernutrition. It is proposed that the emphasis should be on initiatives that improve maternal education and nutrition, child food security, boost household wealth index, and enhance mothers' access to health care. The study strongly recommends that the governments of Bangladesh and India invest financially in preventing child malnutrition, which will contribute to achieving the first four SDGs.

## Introduction

Child malnutrition has emerged as a major public health issue in low-and middle-income countries (LMICs) due to its strong association with child mortality [1, 2]. UNICEF, WHO and World Bank Group jointly estimate that 149.2 million children under the age of 5 are stunted and 45.4 million are wasted globally in 2020, with Asian countries accounting for 79 million stunted and 31.9 million wasted [3]. Bangladesh and India are no exception as LMICs in South Asia. BDHS, 2017–18 estimates, about 30.8% and 8.4% of under-five children are stunted and wasted, respectively [4]. According to the NFHS-4, 2015–16 in India, 38% of under-five children are stunted and 21% are wasted [5].

Bengalis are one of the major populations in South Asia, spread across the world. The partition of India in 1947 is a major event in the division of the Bengalis. Since the independence of Bangladesh in 1971, they have been mostly concentrated in two countries, namely Bangladesh and India. Over 241 million Bengalis lived in these two countries in 2011, with 144,043,697 in Bangladesh and 97,237,669 in India, according to the 2011 census [6, 7]. In India, Bengalis are the country's second-most linguistic community (8.03%), with the majority of them residing in Tripura and West Bengal [7]. Despite their religious diversity, the Bengalis have historically shared a common culture, as they are descendants of the Indo-Aryan branch of the Indo-European linguistic family. Also, they share a similar environment in these two countries in terms of ecology, food habits and socioeconomic background.

Nutritional status is not just hereditary or genetically regulated, but also influenced by other factors (non-genetic) such as dietary habits, socioeconomic circumstances, and environmental factors, hence it is referred to as a multifactorial trait [8, 9]. Malnutrition can be reduced by controlling those non-genetic factors so that the heredity cannot be controlled in general [10, 11]. Many studies have found that child nutrition is affected by a variety of factors, like socioeconomic, demographic, dietary, and maternal factors in nations such as China [12], Iran [13], Ethiopia [14], Nepal [15], Vietnam [16] and many others.

Various studies have been conducted on the nutritional status of Bengali children in Bangladesh and India [17–19]. Most of these studies focus on the nutritional status of certain indicators, including stunting, wasting, and underweight, as well as the factors that influence them. In this perspective, the CIAF is a better indicator of child malnutrition since it expresses all undernutrition indicators in a unified manner [20]. However, no combined study on CIAF in children under 5 has been found in Bengalis of Bangladesh and India. A comparative study

on the nutritional status of these children will help in the development of nutrition development programs for both countries.

The present study intends to compare the CIAF of under 5 Bengali children between India and Bangladesh, as well as to examine the effect of maternal nutrition, socio-demographic and birth-related factors on anthropometric failure. Such information would be relevant for preventing and controlling child malnutrition in the Bengali population and for determining what steps should be taken between the two countries.

## Methods

### Data source

Unit level data for this study was extracted from the BDHS, 2017–2018 in Bangladesh and the NFHS-4, 2015–2016 in India, both of which were publicly available at https://dhsprogram.com/Data/. The NFHS and BDHS were a nationwide survey of a representative sample of households that used standardized questionnaires, sample designs, and field techniques in accordance with Demographic and Health Surveys (DHS) guidelines. All survey protocols for each country were approved by a recognized nodal organization, including the National Institute of Population Research and Training (NIPORT) for the BDHS and the International Institute for Population Sciences (IIPS) for the NFHS-4.

The sample size of the survey consisted of 19457 households in Bangladesh, and 601509 in India. Bengali is the official language in the Indian states of West Bengal and Tripura, located in the western and eastern parts of Bangladesh, respectively. State Tripura and West Bengal were selected as the representative of the Bengali population in India, including 4510 households in Tripura and 15327 in West Bengal. A total of 14055 under 5 children were selected for statistical analysis, with data inclusion and exclusion represented in the flow chart in Fig 1.

### Unit level study variables

**Outcome variable.**   The outcome variable of this study was CIAF, which was estimated using nutritional parameters of stunting, wasting, and underweight under 5 children. The CIAF variable was generated using the definition of anthropometric failure [21]. Dichotomous category of CIAF was applied for data analysis based on the recommendation of [22, 23]. The children with appropriate height and weight for their age (z-score >–2 SD) were classified as having 'no anthropometric failure' (NAF = Group A, Table 1), while those with z-score $\leq$–2 SD were classified as having 'anthropometric failure' (AF = $\sum$ Group B, C, D, E, F, Y). The CIAF classification according to [21] is given in Table 1.

**Explanatory variables.**   To evaluate the determinants of anthropogenic failure (AF), the following variables were included in the study based on previous literature as well as our descriptive findings, as they were common in both countries: household socioeconomic factors, maternal factors, and child factors. Exposure variables were classified using the classifications described in the articles [15, 24–28]. The variables of household socioeconomic factors include, place of residence (urban, rural), religion (Hindu, Islam, Christian & Buddhist), household size (up to three, four, five, more than five), and wealth index (poorest, poorer, middle, richer, richest). In religion, Christians and Buddhists were merged due to low frequency. Maternal factors include body mass index (BMI), which classified based on WHO (2004) recommended cutoff, <18.5 for undernutrition, 18.5 to 24.9 for normal, 25 to 29.9 for overweight, and $\geq$ 30 for obesity. Age of mother (15–17, 18–34, 35–49 years), education of mother (no education, primary, secondary, higher), mode of delivery (vaginal, caesarean), and antenatal care visits (no antenatal visits, one time, two times, three times, four times & above). The

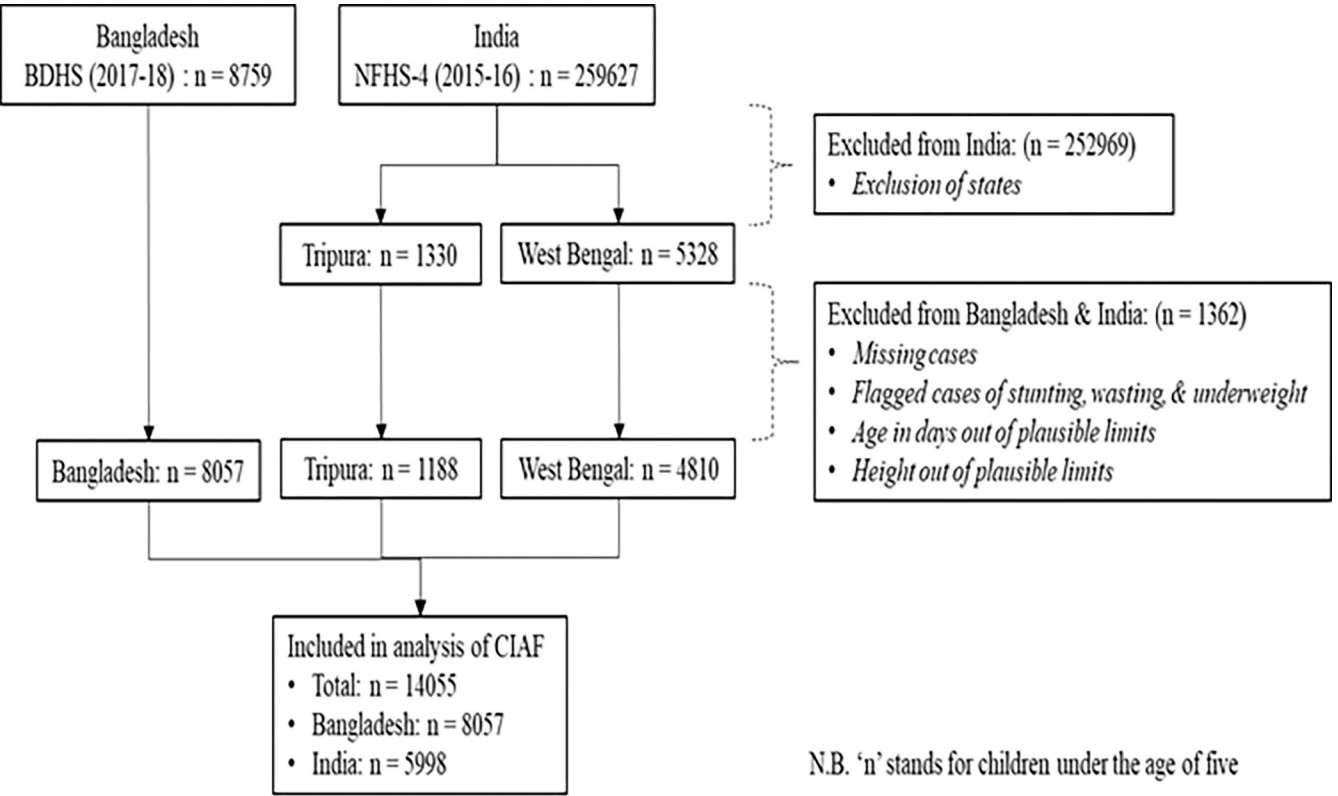

**Fig 1. Sample selection procedure for analysis.**

variables of child factors include, age of children (0–23, 24–59 months), sex of children (boys, girls), number of living children (1–2, >2 children), birth order number (first, second, third, fourth & more), and place of delivery (institution, home). The child's age was classified by breastfeeding, 0–23 months for the breastfeeding period and 24–59 months for the post-breastfeeding period. Some data were further excluded variable wise from the explanatory factors; such as in mother's BMI, 41 data were missing; in religion, category of no religion, others, and Sikh were excluded (151 data) for low frequency; 3077 data of place of delivery (in Bangladesh), 3081 data of mode of delivery (in Bangladesh), 4220 data of antenatal care visits (in both) were excluded for missing.

**Table 1. Classification of composite index of anthropometric failure among under-five children.**

| Group | Description | Wasting | Stunting | Underweight |
|---|---|---|---|---|
| A | No failure | No | No | No |
| B | Wasting only | Yes | No | No |
| C | Wasting and Underweight | Yes | No | Yes |
| D | Wasting, Stunting, and Underweight | Yes | Yes | Yes |
| E | Stunting and Underweight | No | Yes | Yes |
| F | Stunting only | No | Yes | No |
| Y | Underweight only | No | No | Yes |

Note: No anthropometric failure (NAF) = A; Anthropometric failure (AF) = ∑ of B, C, D, E, F, Y

**Statistical analysis.** Descriptive statistics was used to calculate mean and standard deviation for quantitative variables, while frequencies and percentages for categorical variables. The contingency table and Chi-square ($\chi^2$) test were applied to examine the association between categorical variables. The differences in CIAF, A to Y between Bangladesh and India were analyzed using proportional Z test. Binary logistic regression model was applied to identify significant predictors of anthropometric failure. The condition of anthropometric failure was selected as the tested category and was given code '1' and no failure was selected as the reference category and was given code '0'. Following a proper multicollinearity test, independent variables were selected, and the variance inflation factor (VIF) was determined to be less than 5. The p-value was considered significant at level 0.05. Data were analyzed by using the Statistical Package for the Social Sciences (SPSS, version 25.0).

**Ethics approval and consent to participate.** The NFHS-4, 2015–2016 and BDHS, 2017–2018 received ethics approval from the Ministry of Health and Family Welfare, India, and Bangladesh respectively. Both the surveys received written consent from each individual in the study. The survey design, ethics statement, respondents' consent, sampling technique, survey instruments, measuring system and quality control have been described elsewhere for BDHS, 2017–2018 [4] and for NFHS-4, 2015–2016 [5].

## Results

Data comprising Bangladesh and India were 57.3% and 42.7%, respectively, which was quite close to the 60.3% and 39.7% of the total population of Bengalis in Bangladesh and India (Tripura and West Bengal), respectively. The mean age of the children was 28.81 (SD 17.58) months in Bangladesh and 29.85 (SD 17.06) months in India. It was found that the prevalence of anthropometric failure (AF) in Indian children (49.1%) was significantly (p<0.01) higher than that of Bangladeshi children (39%) (Table 2). Also, the prevalence of wasting and underweight in Indian children (8.9%) was significantly (p<0.01) higher than Bangladeshi children (3.0%). AF was found to be more prevalent in boys in Bangladesh, while in India it was more prevalent in girls (Fig 2). However, CIAF revealed that the percentage of groups E (stunting and underweight) and F (stunting only) were higher than in other categories. As a result, stunting (low height for age) and underweight (low weight for age) among Bengali children have become a common public concern in both countries. The distribution of CIAF among under-five children in Bangladesh and India are shown in Fig 3.

**Table 2. Difference in composite index of anthropometric failure of under-five children between Bangladesh and India.**

| Nutritional condition | CIAF Groups | Bangladesh | India | z-value |
|---|---|---|---|---|
| | | n (%) | n (%) | |
| No anthropometric failure | A | 4914 (61.0) | 3053 (50.9) | 8.86** |
| Anthropometric failure (Σ of B, C, D, E, F, Y) | | 3143 (39.0) | 2945 (49.1) | 7.94** |
| Wasting only | B | 199 (2.5) | 315 (5.3) | 1.54[n] |
| Wasting and Underweight | C | 243 (3.0) | 536 (8.9) | 2.98** |
| Wasting, Stunting, Underweight | D | 225 (2.8) | 363 (6.1) | 1.81[n] |
| Stunting and Underweight | E | 1098 (13.6) | 834 (13.9) | 0.19[n] |
| Stunting only | F | 1138 (14.1) | 728 (12.1) | 1.24[n] |
| Underweight only | Y | 240 (3.0) | 169 (2.8) | 0.44[n] |

Note

**: 1% level of significance (p<0.01) and n: insignificance (p>0.05).

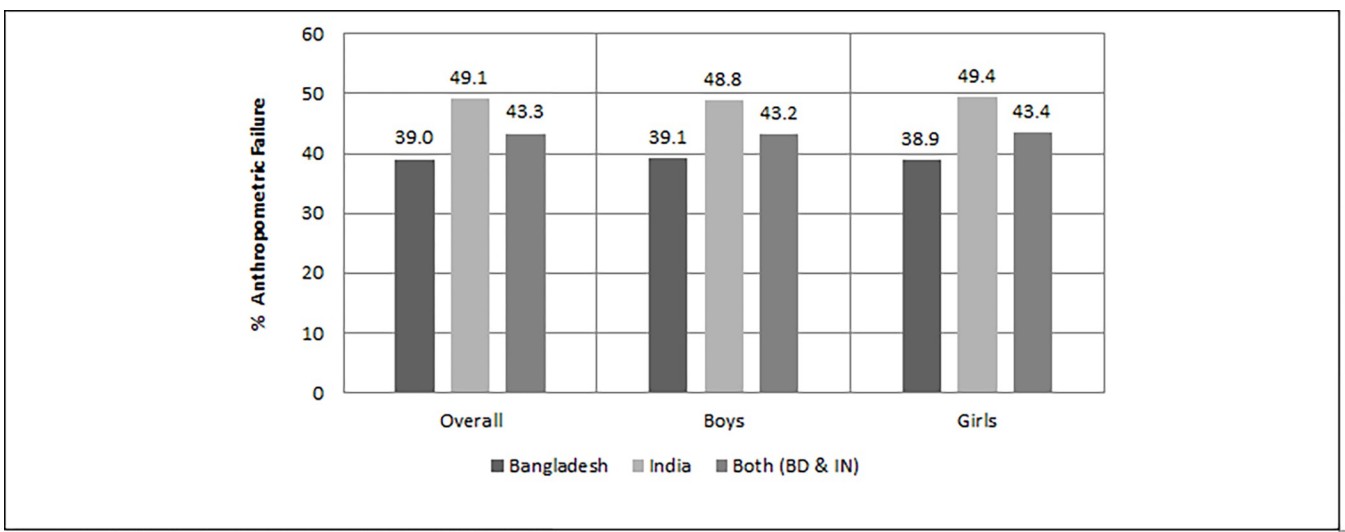

**Fig 2. Prevalence of anthropometric failure among under 5 Bengali children.**

## Distribution of anthropometric failure by explanatory factors

Table 3 shows that the frequency of AF differed significantly ($\chi^2$, p <0.05) from each explanatory factors, except for the sex of children, age of mother, religion in Bangladesh, and household size. Now, AF denoted by 'failure'. Rural areas were observed to have a higher concentration of failure. Apart from India and Bangladesh, failure was more common in Islam, while failure from these two countries together was more common among Hindus. Children from five-member of households and the household with more than two children

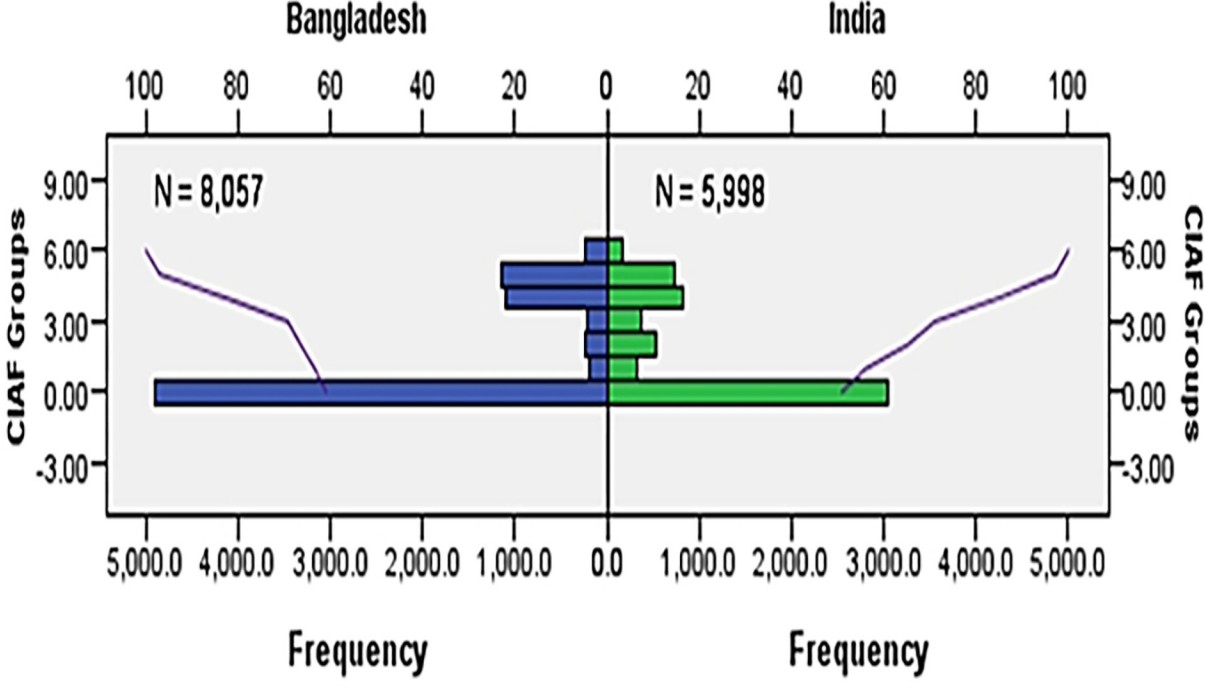

**Fig 3. The distribution of CIAF among under 5 children of Bangladesh and India.**

have a higher rate of failure. Children with a lower wealth index family have a higher rate of failure. The frequency of the nutritional indicators were significantly different with the mother's BMI, failure was more common in underweight mothers. In terms of child's sex and maternal age, failure was distributed almost equally and no significant difference found with its counterpart. Comparatively more failure children were found in less educated mothers, and mothers who did not visit to doctor for antenatal care during their pregnancy. Failure was higher in the post-breastfeeding period than in the breastfeeding period, as seen by the age of children. Birth order number indicates a continuous increasing of failure in respect of increase of birth order number. Failure was more common in children born at home than institutions, also in vaginal birth than caesarean birth (Table 3).

## Effects of explanatory factors on anthropometric failure

The effect of explanatory factors on AF in under-five years Bengali children were shown in Table 4 using binary logistic regression. The model fits the data well, according to the Omnibus chi-square, for Bangladesh 290.53, India 322.87, and both (Bangladesh and India) 620.88, with a significant level p $<$0.01. Each regression models were significant (p$<$0.01), with the correct percentage of prediction for Bangladesh, India, and both being 63.4%, 60.5%, 61.4%, respectively.

**Table 3. Association between anthropometric failure and different selected explanatory factors.**

| Explanatory factors | Anthropometric failure | | |
|---|---|---|---|
| | **Bangladesh** | **India** | **Bangladesh and India (both)** |
| Place of residence | | | |
| Urban | 941 (34.1) | 591 (42.4) | 1532 (36.9) |
| Rural | 2202 (41.6) | 2354 (51.1) | 4556 (46.0) |
| $\chi^2$ (p value) | 42.16 (<0.001) | 32.98 (<0.001) | 99.16 (<0.001) |
| Religion | | | |
| Hindu | 237 (37.1) | 1906 (46.9) | 2143 (45.6) |
| Islam | 2888 (39.2) | 876 (53.8) | 3764 (41.8) |
| Christian & Buddhist | 18 (36.7) | 70 (45.2) | 88 (43.1) |
| $\chi^2$ (p value) | 1.13 (0.568) | 23.04 (<0.001) | 17.66 (<0.001) |
| Household size | | | |
| Up to three | 345 (37.0) | 345 (48.0) | 690 (41.8) |
| Four | 609 (37.6) | 644 (46.8) | 1253 (41.9) |
| Five | 666 (40.8) | 641 (50.4) | 1307 (45.0) |
| More than five | 1523 (39.3) | 1315 (49.9) | 2838 (43.6) |
| $\chi^2$ (p value) | 5.27 (0.153) | 4.83 (0.185) | 7.87 (0.049) |
| Wealth index | | | |
| Poorest | 885 (49.6) | 959 (60.6) | 1844 (54.8) |
| Poorer | 744 (45.9) | 1098 (51.3) | 1842 (48.9) |
| Middle | 546 (37.5) | 535 (44.5) | 1081 (40.7) |
| Richer | 578 (36.2) | 275 (34.9) | 853 (35.8) |
| Richest | 390 (24.4) | 78 (27.4) | 468 (24.8) |
| $\chi^2$ (p value) | 267.35 (<0.001) | 214.43 (<0.001) | 553.52 (<0.001) |
| Mother's BMI | | | |
| Underweight | 622 (52.4) | 945 (60.1) | 1567 (56.8) |
| Normal | 1907 (40.0) | 1740 (47.3) | 3647 (43.2) |
| Overweight | 509 (30.4) | 220 (36.7) | 729 (32.1) |

*(Continued)*

**Table 3.** (Continued)

| Explanatory factors | Anthropometric failure | | |
|---|---|---|---|
| | Bangladesh | India | Bangladesh and India (both) |
| Obese | 95 (23.5) | 29 (22.5) | 124 (23.3) |
| $\chi^2$ (p value) | 183.51 (<0.001) | 154.31 (<0.001) | 407.52 (<0.001) |
| Age of mother | | | |
| 15–17 years | 404 (40.6) | 229 (46.9) | 633 (42.7) |
| 18–34 years | 2473 (38.5) | 2529 (49.2) | 5002 (43.3) |
| 35–49 years | 266 (41.3) | 187 (51.2) | 453 (44.9) |
| $\chi^2$ (p value) | 3.02 (0.219) | 1.59 (0.451) | 1.30 (0.521) |
| Education of mother | | | |
| No education | 311 (54.5) | 706 (62.9) | 1017 (60.1) |
| Primary | 1096 (47.1) | 694 (54.9) | 1790 (49.8) |
| Secondary | 1443 (37.9) | 1441 (44.1) | 2884 (40.8) |
| Higher | 293 (21.7) | 104 (30.0) | 397 (23.4) |
| $\chi^2$ (p value) | 293.38 (<0.001) | 186.27 (<0.001) | 549.39 (<0.001) |
| Mode of delivery | | | |
| Vaginal | 1425 (43.0) | 2486 (52.5) | 3911 (48.6) |
| Caesarean | 494 (29.8) | 459 (36.4) | 953 (32.6) |
| $\chi^2$ (p value) | 81.13 (<0.001) | 103.62 (<0.001) | 220.69 (<0.001) |
| Antenatal care visits | | | |
| No antenatal visits | 196 (50.5) | 261 (56.1) | 457 (53.6) |
| One time | 259 (43.5) | 47 (49.0) | 306 (44.2) |
| Two times | 321 (42.7) | 139 (52.1) | 460 (45.2) |
| Three times | 275 (37.6) | 248 (48.7) | 523 (42.2) |
| Four times & above | 772 (33.4) | 1708 (45.9) | 2480 (41.1) |
| $\chi^2$ (p value) | 60.80 (<0.001) | 20.43 (<0.001) | 50.43 (<0.001) |
| Age of children | | | |
| 0–23 months | 1204 (35.5) | 1108 (47.3) | 2312 (40.3) |
| 24–59 months | 1939 (41.5) | 1837 (50.3) | 3776 (45.4) |
| $\chi^2$ (p value) | 30.02 (<0.001) | 5.16 (0.023) | 35.37 (<0.001) |
| Sex of children | | | |
| Boys | 1644 (39.1) | 1495 (48.8) | 3139 (43.2) |
| Girls | 1499 (38.9) | 1450 (49.4) | 2949 (43.4) |
| $\chi^2$ (p value) | 0.05 (0.826) | 0.19 (0.663) | 0.06 (0.800) |
| Number of living children | | | |
| 1–2 children | 2106 (36.6) | 2238 (46.3) | 4344 (41.0) |
| > 2 children | 1037 (45.0) | 707 (60.7) | 1744 (50.3) |
| $\chi^2$ (p value) | 49.39 (<0.001) | 78.29 (<0.001) | 91.86 (<0.001) |
| Birth order number | | | |
| First | 1113 (36.1) | 1327 (44.7) | 2440 (40.4) |
| Second | 974 (37.1) | 985 (50.9) | 1959 (43.0) |
| Third | 557 (41.1) | 351 (54.4) | 908 (45.4) |
| Fourth & more | 499 (49.9) | 282 (62.7) | 781 (53.9) |
| $\chi^2$ (p value) | 67.36 (<0.001) | 65.38 (<0.001) | 91.52 (<0.001) |
| Place of delivery | | | |
| Institution | 818 (32.7) | 2121 (46.3) | 2939 (41.5) |
| Home | 1102 (44.4) | 824 (58.3) | 1926 (49.4) |
| $\chi^2$ (p value) | 71.37 (<0.001) | 62.32 (<0.001) | 64.14 (<0.001) |

**Factors associated with AF in Bangladesh.** AF was more common in children from poor wealth index families. Maternal undernutrition had a positive effect on child malnutrition, indicating that children of underweight mothers were 1.46 times more likely to suffer from AF (AOR: 1.46, 95% CI: 1.24, 1.72). Maternal education had a negative effect on child malnutrition, with mothers who were non-educated or less educated their children mostly suffering from AF compared to higher educated mothers. A vaginal birth children were 19% more likely to develop AF (AOR: 0.81, 95% CI: 0.67, 0.98). Mothers who had never received antenatal care were 1.28 times more likely to have AF in their children (AOR: 1.28, 95% CI: 1.00, 1.62). Age of the children indicates, at the end of breastfeeding children were 1.56 times more likely to suffer from AF than breastfeeding children (AOR: 1.56, 95% CI: 1.37, 1.78). Boys were 13% more likely to suffer from AF than girls (AOR: 0.87, 95% CI: 0.77, 0.98) (Table 4).

**Factors associated with AF in India.** It was found that children living in a large family (more than five members) had 26% more chance to have AF compared to children living in small family (up to three members) (AOR: 0.74, 95% CI: 0.60, 0.92). The wealth index indicated that the poorest family was twice (AOR: 2.08, 95% CI: 1.45, 2.99) as likely to have an AF child as the richest family. Maternal malnutrition was found to be a significant predictor of child malnutrition, showing that underweight mothers were more likely (AOR: 1.44, 95% CI: 1.25, 1.65), while overweight and obese mothers were less likely to have AF children as compared to healthy mothers (normal BMI). Lower maternal education was found to be positively associated with AF, as non-educated mothers were 1.59 times more likely to have children with AF (AOR: 1.59, 95% CI: 1.14, 2.21). We found that vaginal birth children had 22% more chance of suffering from AF compared to children who were born through the caesarean section (AOR: 0.78, 95% CI: 0.67, 0.92). Birth order indicated that a second child was 1.23 times more likely to suffer from AF than the first child (AOR: 1.23, 95% CI: 1.06, 1.43) (Table 4).

**Factors associated with AF in both Bangladesh and India.** Family size was a common predictor, indicating that children belonging to a large family (more than five members) had 17% more likely to suffer from AF than children living in a small family with up to three members (AOR: 0.83, 95% CI: 0.71, 0.97). Wealth index indicated that children from poor families were most likely suffering from AF. Maternal BMI was found to be a common predictor in both counties, indicating underweight mothers were more likely (AOR: 1.48, 95% CI: 1.33, 1.65), while overweight and obese mothers were less likely to have AF children as compared to normal BMI mothers. Lower maternal education was positively associated with AF, with lower or non-educated mothers having a higher probability than higher educated mothers. We found that vaginal birth children were 24% more likely to have AF compared to children who was born through caesarean section (AOR: 0.76, 95% CI: 0.68, 0.86). Child age was also a common predictor of AF, with children 1.32 times more likely to suffer from AF after the end of breastfeeding (AOR: 1.32, 95% CI: 1.21, 1.44) (Table 4).

## Discussion

This study aims to explore the prevalence of anthropometric failure among Bengali children under the age of 5 in Bangladesh and India, as well as evaluate the impact of maternal nutrition, socio-demographic, and birth-related factors on AF using DHS data. The overall prevalence of AF was higher in India (49.1%), although outbreaks appear in CIAF's group F (stunting only) and group E (stunting and underweight) among Bengali children in both countries, comprising 71% for Bangladesh and 53% for India in terms of total AF. It was found that AF among Bengali children in Bangladesh and India had been associated with maternal nutrition, socioeconomic demographic variables, and birth-related factors. Some of the factors

**Table 4. Effect of socioeconomic, demographic and anthropometric factors on anthropometric failure among under-five Bengali children.**

| Explanatory Factors | | Bangladesh | | India | | Bangladesh and India (both) | |
|---|---|---|---|---|---|---|---|
| | | AOR (95% CI) | p-value | AOR (95% CI) | p-value | AOR (95% CI) | p-value |
| Place of residence | | | | | | | |
| | *Urban* | Reference | | Reference | | Reference | |
| | Rural | 0.99 (0.86, 1.14) | 0.877 | 0.99 (0.85, 1.16) | 0.906 | 1.00 (0.90, 1.11) | 0.991 |
| Religion | | | | | | | |
| | *Hindu* | Reference | | Reference | | Reference | |
| | Islam | 1.19 (0.94, 1.50) | 0.149 | 1.11 (0.97, 1.28) | 0.138 | 1.02 (0.93, 1.13) | 0.674 |
| | Christian & Buddhist | 1.03 (0.44, 2.43) | 0.941 | 0.94 (0.64, 1.36) | 0.730 | 0.92 (0.66, 1.30) | 0.650 |
| Household size | | | | | | | |
| | *More than five* | Reference | | Reference | | Reference | |
| | Upto three | 0.95 (0.74, 1.20) | 0.640 | 0.74 (0.60, 0.92) | 0.007 | 0.83 (0.71, 0.97) | 0.019 |
| | Four | 1.07 (0.85, 1.35) | 0.578 | 0.86 (0.70, 1.07) | 0.175 | 0.95 (0.81, 1.11) | 0.475 |
| | Five | 0.97 (0.79, 1.19) | 0.749 | 0.90 (0.75, 1.09) | 0.290 | 0.93 (0.81, 1.07) | 0.289 |
| Wealth index | | | | | | | |
| | *Richest* | Reference | | Reference | | Reference | |
| | Poorest | 1.50 (1.19, 1.91) | 0.001 | 2.08 (1.45, 2.99) | <0.001 | 1.73 (1.43, 2.10) | <0.001 |
| | Poorer | 1.34 (1.06, 1.69) | 0.014 | 1.56 (1.11, 2.21) | 0.011 | 1.41 (1.17, 1.69) | <0.001 |
| | Middle | 1.18 (0.94, 1.47) | 0.158 | 1.48 (1.06, 2.07) | 0.022 | 1.28 (1.07, 1.52) | 0.007 |
| | Richer | 1.11 (0.89, 1.37) | 0.359 | 1.22 (0.88, 1.70) | 0.238 | 1.11 (0.93, 1.32) | 0.240 |
| Mother's BMI | | | | | | | |
| | *Normal* | Reference | | Reference | | Reference | |
| | Underweight | 1.46 (1.24, 1.72) | <0.001 | 1.44 (1.25, 1.65) | <0.001 | 1.48 (1.33, 1.65) | <0.001 |
| | Overweight | 0.89 (0.75, 1.06) | 0.184 | 0.80 (0.65, 0.97) | 0.025 | 0.84 (0.74, 0.96) | 0.008 |
| | Obese | 0.71 (0.50, 1.00) | 0.052 | 0.40 (0.25, 0.64) | <0.001 | 0.56 (0.43, 0.74) | <0.001 |
| Age of mother | | | | | | | |
| | *35–49 years* | Reference | | Reference | | Reference | |
| | 15–17 years | 0.94 (0.67, 1.32) | 0.726 | 1.03 (0.74, 1.45) | 0.854 | 1.02 (0.80, 1.28) | 0.900 |
| | 18–34 years | 1.02 (0.77, 1.35) | 0.913 | 1.08 (0.83, 1.40) | 0.564 | 1.06 (0.88, 1.28) | 0.561 |
| Education of mother | | | | | | | |
| | *Higher* | Reference | | Reference | | Reference | |
| | No education | 2.21 (1.61, 3.04) | <0.001 | 1.59 (1.14, 2.21) | 0.006 | 2.22 (1.80, 2.73) | <0.001 |
| | Primary | 1.91 (1.52, 2.39) | <0.001 | 1.36 (1.00, 1.87) | 0.053 | 1.73 (1.45, 2.07) | <0.001 |
| | Secondary | 1.75 (1.44, 2.14) | <0.001 | 1.12 (0.84, 1.48) | 0.440 | 1.51 (1.29, 1.77) | <0.001 |
| Mode of delivery | | | | | | | |
| | *Vaginal* | Reference | | Reference | | Reference | |
| | Caesarean | 0.81 (0.67, 0.98) | 0.029 | 0.78 (0.67, 0.92) | 0.002 | 0.76 (0.68, 0.86) | <0.001 |
| Antenatal care visits | | | | | | | |
| | *Four times & above* | Reference | | Reference | | Reference | |
| | No antenatal visits | 1.28 (1.00, 1.62) | 0.047 | 0.99 (0.80, 1.23) | 0.947 | 1.09 (0.93, 1.28) | 0.292 |
| | One time | 1.12 (0.92, 1.37) | 0.254 | 1.04 (0.68, 1.59) | 0.853 | 0.98 (0.83, 1.16) | 0.810 |
| | Two times | 1.18 (0.99, 1.41) | 0.069 | 1.02 (0.78, 1.32) | 0.906 | 1.06 (0.92, 1.22) | 0.459 |
| | Three times | 1.03 (0.86, 1.24) | 0.726 | 0.94 (0.77, 1.14) | 0.513 | 0.95 (0.84, 1.08) | 0.452 |
| Age of children | | | | | | | |
| | *0–23 months* | Reference | | Reference | | Reference | |
| | 24–59 months | 1.56 (1.37, 1.78) | <0.001 | 1.09 (0.96, 1.23) | 0.184 | 1.32 (1.21, 1.44) | <0.001 |
| Sex of children | | | | | | | |
| | *Boys* | Reference | | Reference | | Reference | |

*(Continued)*

**Table 4.** (Continued)

| Explanatory Factors | | Bangladesh | | India | | Bangladesh and India (both) | |
|---|---|---|---|---|---|---|---|
| | | AOR (95% CI) | p-value | AOR (95% CI) | p-value | AOR (95% CI) | p-value |
| | Girls | 0.87 (0.77, 0.98) | 0.020 | 0.99 (0.89, 1.12) | 0.923 | 0.93 (0.86, 1.02) | 0.106 |
| Number of Living children | | | | | | | |
| | ≤ 2 children | Reference | | Reference | | Reference | |
| | > 2 children | 0.71 (0.49, 1.02) | 0.064 | 1.29 (0.84, 1.98) | 0.247 | 0.93 (0.71, 1.23) | 0.624 |
| Birth order number | | | | | | | |
| | First | Reference | | Reference | | Reference | |
| | Second | 0.86 (0.72, 1.02) | 0.077 | 1.23 (1.06, 1.43) | 0.007 | 1.05 (0.94, 1.17) | 0.390 |
| | Third | 1.16 (0.81, 1.67) | 0.424 | 0.96 (0.64, 1.44) | 0.840 | 1.09 (0.83, 1.43) | 0.521 |
| | Fourth & more | 1.30 (0.86, 1.97) | 0.221 | 1.16 (0.72, 1.89) | 0.542 | 1.24 (0.90, 1.69) | 0.189 |
| Place of delivery | | | | | | | |
| | Institution | Reference | | Reference | | Reference | |
| | Home | 1.04 (0.87, 1.24) | 0.675 | 1.04 (0.88, 1.22) | 0.655 | 0.98 (0.88, 1.10) | 0.758 |

were common in both countries, while others varied. Now all those factors were discussed in detail:

Maternal undernutrition has been identified as a primary factor of AF in children of both countries. A similar finding was reported in Ethiopia, where the study found that mothers who were normal or overweight have a lower risk of having undernourished (or CIAF) children [14].

Child age was not a significant factor in India, particularly over the age of 23 months, but it was in Bangladesh. That indicates, children in India receive adequate nutrients after they quit breastfeeding, whereas children in Bangladesh did not get as much. As a result, if children in Bangladesh after quit breastfeeding they were more likely to develop AF.

In India, there was no nutritional difference between boys and girls in terms of CIAF, whereas AF was lower among girls in Bangladesh than boys.

Maternal education has been identified as a common factor of child malnutrition, with lower educated mothers had a higher risk of having AF than higher educated mothers.

The children from poor families had a higher risk of AF than those from richer ones. Vollmer and colleagues used DHS data across 39 countries to estimate the prevalence of CIAF by focusing on maternal education and wealth index, indicating that child nutrition co-exists with lower maternal education and poor wealth index [29].

Caesarean delivery had a lower chance of AF, a condition that was indirectly related to maternal health. In general caesarean delivery is more common in obese mothers when vaginal delivery is not possible. This suggests that caesarean delivery does not directly reduce child malnutrition, but it is made possible by improved maternal health.

AF was less prevalent among children of households with only three members in India. In most cases, households with three members have only one child, the parents and their single child. Child undernutrition may be reduced in this instance due to proper food distribution and child care.

In India, first children were more prone to undernutrition. In the case of the first child, the mother's education and antenatal care play an important role in child health. It can be noticed that the rate of non-education and rejection of antenatal care among mothers in India was comparatively higher than in Bangladesh.

In Bangladesh, mothers who did not have a single antenatal visit had a higher incidence of AF among their children. Other studies abroad in Latin America including Bolivia, Colombia,

and Peru [30], Nepal [31], Yemen [32], and Thailand [33] have shown the coexistence of antenatal care and child undernutrition.

## Conclusions

In conclusion, anthropometric failure is a community burden among under-five Bengali children in both Bangladesh and India. Among Bengalis, 5 out of every 10 children in India and 4 children in Bangladesh suffer from anthropometric failure, there is a significant difference between the two countries in terms of AF. Children have a higher prevalence of stunting and underweight in both countries. Anthropometric failure coexists with maternal health, socio-demographic, and birth-related covariates. Maternal undernutrition, lack of maternal education, lack of antenatal care, poor wealth index, and an insufficient supply of nutrients after the end of the breastfeeding period all contribute to an increase in the incidence of AF in both countries. Intensive lifestyle improvement should be implemented early to reduce child undernutrition. Public health initiatives aimed at improving maternal health at the population level are expected to have a positive impact on lowering child undernutrition. Bengalis need to improve their economy holistically so that they can purchase sufficient nutritious food. Otherwise, the government of both countries must secure an adequate supply of nutritious food and ensure that food distribution schemes are properly implemented. Simultaneously, mothers should be educated and provide proper knowledge on child health, which is expected to have a greater impact on reducing the burden of AF in Bangladesh and India.

## Acknowledgments

The authors would like to thank the IRB for allowing the use of DHS data for the study. Authors gratefully thank to the Health Research Group of the Department of Statistics, Rajshahi University, Bangladesh, and Rashidul Alam Mahumud, Informatics and Economic Research, University of Southern Queensland, Australia for providing important suggestions for this study.

## Author Contributions

**Conceptualization:** Ramendra Nath Kundu, Md. Golam Hossain, Premananda Bharati.

**Data curation:** Md. Kamal Pasa, Md. Sabiruzzaman.

**Formal analysis:** Ramendra Nath Kundu, Md. Golam Hossain, Md. Ahshanul Haque, Md. Monimul Huq.

**Investigation:** Subir Biswas.

**Methodology:** Ramendra Nath Kundu, Md. Golam Hossain, Md. Ahshanul Haque, Md. Monimul Huq, Md. Kamal Pasa, Md. Sabiruzzaman.

**Resources:** Md. Golam Hossain, Premananda Bharati.

**Supervision:** Md. Golam Hossain, Premananda Bharati.

**Validation:** Md. Ahshanul Haque, Subir Biswas, Md. Monimul Huq, Md. Kamal Pasa, Md. Sabiruzzaman.

**Visualization:** Md. Monimul Huq, Md. Sabiruzzaman.

**Writing – original draft:** Ramendra Nath Kundu.

**Writing – review & editing:** Ramendra Nath Kundu, Md. Golam Hossain, Md. Ahshanul Haque, Subir Biswas, Md. Monimul Huq, Md. Kamal Pasa, Md. Sabiruzzaman, Premananda Bharati.

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
