## [Decision Letter · Decision Letter 0]

17 Feb 2022

PONE-D-21-41052Factor associated with anthropometric failure among under-five Bengali children: a comparative study between Bangladesh and IndiaPLOS ONE

Dear Dr. Hossain,

Thank you for submitting your manuscript to PLOS ONE. After careful consideration, we feel that it has merit but does not fully meet PLOS ONE’s publication criteria as it currently stands. Therefore, we invite you to submit a revised version of the manuscript that addresses the points raised during the review process.

We look forward to receiving your revised manuscript.

Kind regards,

Lai Kuan Lee

Academic Editor

PLOS ONE

Journal Requirements:

3. PLOS ONE does not copy edit accepted manuscripts (https://journals.plos.org/plosone/s/criteria-for-publication#loc-5). To that effect, please ensure that your submission is free of typos and grammatical errors.

Additional Editor Comments:

Dear Dr. Golam Hossain,

The reviewers are now commenting on your manuscript. Please pay special attention to the following points if you wish to submit a revised version of the manuscript:

- Referencing list: The format is not justified.

- Conclusion: We wish to see some suggestions on the policy drafting directions

Reviewers' comments:

Reviewer's Responses to Questions

**Comments to the Author**

1. Is the manuscript technically sound, and do the data support the conclusions?

Reviewer #1: Partly

Reviewer #2: Yes

2. Has the statistical analysis been performed appropriately and rigorously? 

Reviewer #1: Yes

Reviewer #2: Yes

3. Have the authors made all data underlying the findings in their manuscript fully available?

Reviewer #1: Yes

Reviewer #2: Yes

4. Is the manuscript presented in an intelligible fashion and written in standard English?

Reviewer #1: No

Reviewer #2: No

5. Review Comments to the Author

Reviewer #1: Title: Factor associated with anthropometric failure among under-five Bengali children: a comparative study between Bangladesh and India

Manuscript Number: PONE-D-21-41052

Comments to the Authors

Dear Authors,

The title is relevant and addresses the public health problem of under-five children. However, the following comments shall be considered in order to improve it. In addition, the entire manuscript requires extensive language editing.

Results

1. Line 165-166 and under table 2, is there any statistically significant difference in the prevalence of anthropometric failure (AF) among under-five children in India and Bangladesh? If yes, please state it clearly?

2. On lines 201, 211 & 221, it is better to say “Factors associated with AF….” instead of saying “Predictors of AF…....”

3. Lines 201-230, please incorporate the AOR with 95% CI at the end of each statement which deals with associated factors.

Discussion

1. Lines 233-240, shall be revised. It would be better to put the summary of the aim and method instead of stating it as such.

2. For better communication and understanding, please state each factor in a paragraph instead of discussing more than one factor in a paragraph.

3. It is better to revise the strength and limitations of the study. If not, better to remove it since it does not seem the strength and limitations of the study.

Conclusions

1. Lines 297-299, the conclusion should be drawn based on the findings of the study. Thus, please remove the statement “Present study will contribute to achieving four SDGs of the 298 United Nations out of seventeen, no poverty, zero hunger, good health and well-being, and 299 quality education”. You have already stated it under the introduction section of the manuscript.

2. The conclusion, abbreviation, and acknowledgment should be written as a chapter heading on a separate line.

References

1. Please cite the references properly. Don’t mix up it.

Figure Legends

Lines 433-436, Figure captions should be self-explanatory. So, please revise it.

Reviewer #2: This manuscript is very good and well organized and constructed. it can be accepted and published, it will help the health professionals in understanding the compensate index of anthropometric failure.

6. PLOS authors have the option to publish the peer review history of their article (what does this mean?). If published, this will include your full peer review and any attached files.

Reviewer #1: No

Reviewer #2: No

---

## [Author Response · Author response to Decision Letter 0]

16 Mar 2022

Response to Reviewers Date: March 11, 2022

Paper Title: Factor associated with anthropometric failure among under-five Bengali children: a comparative study between Bangladesh and India

Journal Name: PLOS ONE

Paper ID: PONE-D-21-41052

Dear Editor

Thank you very much for providing you and reviewers’ insightful remarks on our manuscript. We have made the necessary changes and revised the manuscript accordingly, and detailed point–by–point corrections are given below:

Review Reports:

Reviewer 1

The title is relevant and addresses the public health problem of under-five children. However, the following comments shall be considered in order to improve it. In addition, the entire manuscript requires extensive language editing.

Response to Reviewer Comments: Thank you very much for your comments/suggestions on our manuscript. We have tried with our best to revise our manuscript accordingly. 

Results

Reviewer Comment # 1: Line 165-166 and under table 2, is there any statistically significant difference in the prevalence of anthropometric failure (AF) among under-five children in India and Bangladesh? If yes, please state it clearly?

Response to Reviewer Comments: Based on your point, we applied Z-proportional test to check if there was a significant difference in the categories of anthropometric failure (AF) between the under-five children of Bangladesh and India, and we noticed that there were significant differences [Line: 165-170; Page: 8-9].

Reviewer Comment # 2: On lines 201, 211 & 221, it is better to say “Factors associated with AF….” instead of saying “Predictors of AF…....”

Response to Reviewer Comments: Thank you for your suggestion. We have checked and made correction.

Reviewer Comment # 3: Lines 201-230, please incorporate the AOR with 95% CI at the end of each statement which deals with associated factors.

Response to Reviewer Comments: Thank you for your suggestion. We have kept the original content unaltered and added AOR with 95% CI to lines 201 to 230. In the case of variables with more than two categories have more than two AOR and 95% CI have to be mentioned, where we noticed that the ease of reading the sentences is reducing. In this situation, we have tried to mention AOR and CI as many as variables allows.

Discussion

Reviewer Comment # 1: Lines 233-240, shall be revised. It would be better to put the summary of the aim and method instead of stating it as such.

Response to Reviewer Comments: Thank you for your suggestion. We have added the summary of the aim and method at the beginning of the discussion and removed the sentences in lines 233-240, as you recommended.

Reviewer Comment # 2: For better communication and understanding, please state each factor in a paragraph instead of discussing more than one factor in a paragraph.

Response to Reviewer Comments: Thank you for your comment. We agree with you, we have described each factor in a separate paragraph. Which made the discussion a lot easier to read.

Reviewer Comment # 3: It is better to revise the strength and limitations of the study. If not, better to remove it since it does not seem the strength and limitations of the study.

Response to Reviewer Comments: Thank you for your suggestion. We have removed the point ‘strength and limitations of the study’ as per your suggestion.

Conclusions

Reviewer Comment # 1: Lines 297-299, the conclusion should be drawn based on the findings of the study. Thus, please remove the statement “Present study will contribute to achieving four SDGs of the 298 United Nations out of seventeen, no poverty, zero hunger, good health and well-being, and 299 quality education”. You have already stated it under the introduction section of the manuscript.

Response to Reviewer Comments: Thank you for your suggestion. The sentence you mentioned has been removed.

Reviewer Comment # 2: The conclusion, abbreviation, and acknowledgment should be written as a chapter heading on a separate line.

Response to Reviewer Comments: Thank you for your suggestion. We have checked and made correction.

References

Reviewer Comment # 1: Please cite the references properly. Don’t mix up it.

Response to Reviewer Comments: Thank you for pointing this out. In order to make some changes in the manuscript, the changes made in the reference case were revised and rearranged.

Figure Legends

Reviewer Comment: Lines 433-436, Figure captions should be self-explanatory. So, please revise it.

Response to Reviewer Comments: Thank you for your comment. We have modified figure’s caption.

Reviewer 2 

Reviewer Comment: This manuscript is very good and well organized and constructed. It can be accepted and published, it will help the health professionals in understanding the compensate index of anthropometric failure.

Response to Reviewer Comments: Thank you very much for your valuable comment. 

We would like to thank the reviewers for the valuable comments. We have revised the documents to the best of our ability, but we will definitely be happy to provide further improvement if there are further clarifications required. 

With best regards

Dr. Md. Golam Hossain

Professor of Health Research Group

Department of Statistics, University of Rajshahi

Rajshahi-6205, Bangladesh

E-mail: hossain95@yahoo.com

---

## [Decision Letter · Decision Letter 1]

22 Jun 2022

PONE-D-21-41052R1Factor associated with anthropometric failure among under-five Bengali children: a comparative study between Bangladesh and IndiaPLOS ONE

Dear Dr. Hossain,

Thank you for submitting your manuscript to PLOS ONE. After careful consideration, we feel that it has merit but does not fully meet PLOS ONE’s publication criteria as it currently stands. Therefore, we invite you to submit a revised version of the manuscript that addresses the points raised during the review process.

The revised version has been partially addressed. Kindly look into the comments again to improve the manuscript.

We look forward to receiving your revised manuscript.

Kind regards,

Lai Kuan Lee

Academic Editor

PLOS ONE

Journal Requirements:

Reviewers' comments:

Reviewer's Responses to Questions

**Comments to the Author**

1. If the authors have adequately addressed your comments raised in a previous round of review and you feel that this manuscript is now acceptable for publication, you may indicate that here to bypass the “Comments to the Author” section, enter your conflict of interest statement in the “Confidential to Editor” section, and submit your "Accept" recommendation.

Reviewer #1: (No Response)

2. Is the manuscript technically sound, and do the data support the conclusions?

Reviewer #1: Yes

3. Has the statistical analysis been performed appropriately and rigorously? 

Reviewer #1: Yes

4. Have the authors made all data underlying the findings in their manuscript fully available?

Reviewer #1: Yes

5. Is the manuscript presented in an intelligible fashion and written in standard English?

Reviewer #1: Yes

6. Review Comments to the Author

Reviewer #1: Dear authors,

You have addressed my comments partially. But, still, I have some concerns.

1. Lines 110-115: Ethical approval and consent to participate shall be stated next to the statistical analysis section.

2. Table and figure captions are not self-explanatory. Still, they need revision. So, please revise them again.

3. In the result section on lines 218-219, 223, 226, 234-235, 240, 246-247; please interpret the AOR findings correctly. When the AOR is less than 1 it should be subtracted from 1 in order to interpret it.

4. Lines 336-349: The references should be cited properly. References 3-8 were structured in Harvard Style. Please structure them in Vancouver style instead of Harvard style.

5. Still the manuscript needs language editing.

7. PLOS authors have the option to publish the peer review history of their article (what does this mean?). If published, this will include your full peer review and any attached files.

Reviewer #1: No

---

## [Author Response · Author response to Decision Letter 1]

26 Jun 2022

Response to Reviewers Date: June 26, 2022

Paper Title: Factor associated with anthropometric failure among under-five Bengali children: a comparative study between Bangladesh and India

Journal Name: PLOS ONE

Paper ID: PONE-D-21-41052R1

Dear Editor

Thank you very much for providing you and reviewers’ insightful remarks on our manuscript. We have made the necessary changes and revised the manuscript accordingly, and detailed point–by–point corrections are given below:

Journal Requirements: 

Response for Journal Requirements: We have checked the reference list and have found some mistakes that have been corrected. 

Review Reports:

Reviewer #1: Dear authors,

You have addressed my comments partially. But, still, I have some concerns.

1. Lines 110-115: Ethical approval and consent to participate shall be stated next to the statistical analysis section.

Response to Reviewer Comments: Thank you very much for your comments/suggestions on our manuscript. According to your suggestion, we have stated “Ethical approval and consent to participate” next to the statistical analysis section [Line: 161-166]. 

2. Table and figure captions are not self-explanatory. Still, they need revision. So, please revise them again.

Response to Reviewer Comments: We have revised tables and figures captions. 

3. In the result section on lines 218-219, 223, 226, 234-235, 240, 246-247; please interpret the AOR findings correctly. When the AOR is less than 1 it should be subtracted from 1 in order to interpret it.

Response to Reviewer Comments: We have interpreted AOR as per your suggestions. 

4. Lines 336-349: The references should be cited properly. References 3-8 were structured in Harvard Style. Please structure them in Vancouver style instead of Harvard style.

Response to Reviewer Comments: Thank you very much for your comments. We have checked and made corrections. 

5. Still the manuscript needs language editing.

Response to Reviewer Comments: We have tried with our best to make grammatical corrections. 

We would like to thank the reviewers for the valuable comments. We have revised the documents to the best of our ability, but we will definitely be happy to provide further improvement if there are further clarifications required. 

With best regards

Dr. Md. Golam Hossain

Professor of Health Research Group

Department of Statistics, University of Rajshahi

Rajshahi-6205, Bangladesh

E-mail: hossain95@yahoo.com

---

## [Decision Letter · Decision Letter 2]

25 Jul 2022

Factor associated with anthropometric failure among under-five Bengali children: a comparative study between Bangladesh and India

PONE-D-21-41052R2

Dear Dr. Hossain,

We’re pleased to inform you that your manuscript has been judged scientifically suitable for publication and will be formally accepted for publication once it meets all outstanding technical requirements.

Kind regards,

Lai Kuan Lee

Academic Editor

PLOS ONE

Additional Editor Comments (optional):

The manuscript is acceptable to be published in its current form.

Reviewers' comments:

Reviewer's Responses to Questions

**Comments to the Author**

1. If the authors have adequately addressed your comments raised in a previous round of review and you feel that this manuscript is now acceptable for publication, you may indicate that here to bypass the “Comments to the Author” section, enter your conflict of interest statement in the “Confidential to Editor” section, and submit your "Accept" recommendation.

Reviewer #1: All comments have been addressed

2. Is the manuscript technically sound, and do the data support the conclusions?

Reviewer #1: Yes

3. Has the statistical analysis been performed appropriately and rigorously? 

Reviewer #1: Yes

4. Have the authors made all data underlying the findings in their manuscript fully available?

Reviewer #1: Yes

5. Is the manuscript presented in an intelligible fashion and written in standard English?

Reviewer #1: Yes

6. Review Comments to the Author

Reviewer #1: Dear authors,

Thank you for addressing my comments. Now, it can be publishable in PLOS ONE journal.

7. PLOS authors have the option to publish the peer review history of their article (what does this mean?). If published, this will include your full peer review and any attached files.

Reviewer #1: No

---

## [Editor Report · Acceptance letter]

28 Jul 2022

PONE-D-21-41052R2 

Factor associated with anthropometric failure among under-five Bengali children: a comparative study between Bangladesh and India 

Dear Dr. Hossain:

I'm pleased to inform you that your manuscript has been deemed suitable for publication in PLOS ONE. Congratulations! Your manuscript is now with our production department. 

Kind regards, 

on behalf of

Dr. Lai Kuan Lee 

Academic Editor

PLOS ONE